# The Relationship between Neuropsychiatric Symptoms and Cognitive Performance in Older Adults with Normal Cognition

**DOI:** 10.3390/medicina58111586

**Published:** 2022-11-03

**Authors:** Ioannis Liampas, Vasileios Siokas, Constantine George Lyketsos, Efthimios Dardiotis

**Affiliations:** 1Department of Neurology, University Hospital of Larissa, Faculty of Medicine, University of Thessaly, 41100 Larissa, Greece; 2Department of Psychiatry and Behavioral Sciences, Johns Hopkins School of Medicine, Baltimore, MD 21205, USA

**Keywords:** hallucinations, delusions, anxiety, depression, apathy

## Abstract

*Background and Objectives*: To explore whether specific Neuropsychiatric Symptoms (NPS) are related to worse performance in particular cognitive domains. *Materials and Methods*: A cross-sectional analysis of the baseline evaluations of older (≥60 years), cognitively unimpaired (CU) participants from the National Alzheimer’s Coordinating Center (NACC) Uniform Data Set was performed. Data were derived from 43 Alzheimer’s Disease Research Centers. Cognitively impaired individuals, participants with psychiatric disorders and/or under treatment with antipsychotic, anxiolytic, sedative, or hypnotic agents were excluded. NPS were assessed using the Neuropsychiatric Inventory Questionnaire. The association of NPS with participants’ performance on episodic memory, semantic memory, language, attention, processing speed and executive function was analysed using an adjusted (considering important demographic and medical factors) multivariate general linear model. *Results*: A total of 7179 CU, older, predominantly female, Caucasian, and well-educated participants were included in the present analysis. Among them, 1856 individuals had one or more NPS. Our analysis revealed that moderate/severe anxiety was related to worse performance on semantic memory, attention and executive function, the presence of hallucinations was linked to worse processing speed and executive function scores, while the presence of elation/euphoria and aberrant motor behaviour were associated with poorer attention and language performance, respectively. In the context of a secondary, exploratory analysis, the presence of moderate/severe delusions was related to worse processing speed and executive function performance. *Conclusions*: The relationship between specific NPS and worse performance in particular cognitive domains could inform the formulation of individualized preventive strategies directed to the ‘‘fortification’’ of specific cognitive functions in CU individuals with NPS.

## 1. Introduction

Neuropsychiatric symptoms (NPS) are prevalent among individuals with mild cognitive impairment (MCI) and dementia [1]. Of note, the frequency and severity of NPS increase in parallel with the severity of cognitive impairment in both conditions [2,3]. In cognitively unimpaired (CU) older individuals, NPS are associated with more rapid cognitive decline and higher risk of incident MCI and dementia [4,5,6]. Of importance, NPS are found in the majority (~60%) of CU individuals who progress to MCI or dementia, often manifesting for some years prior to the diagnosis of any cognitive disorder [7]. Therefore, the identification of NPS in CU individuals ought to increase clinical vigilance regarding the development of incident cognitive impairment.

Unfortunately, the management of NPS has not proven sufficient for altering the course of cognitive decline in older adults. Many traditional medicinal options for NPS, including antipsychotics, benzodiazepines and tricyclic antidepressants, have a well-established detrimental impact on cognition themselves [8,9,10]. Moreover, newer psychotropic agents and namely selective serotonin reuptake inhibitors appear ineffective in improving cognitive outcomes with occasional evidence suggesting that they may even exert a deleterious effect on cognition, as well [11,12,13]. In parallel, there is only sparse and conflicting evidence indicating that psychological-behavioural interventions may improve cognitive performance or alter the course of cognitive decline in older adults with NPS [14,15,16,17,18].

Given the suboptimal efficacy of the available therapeutic options targeting NPS on cognitive outcomes, the implementation of preventive interventions targeting cognition to slow down or preclude cognitive decline, MCI as well as dementia development in CU older adults with NPS is probably warranted. However, it is uncertain which specific cognitive domains (if any) are affected by the presence of particular NPS. Previous researchers have mostly focused on composite neuropsychiatric and cognitive measures while the specific associations between individual NPS and cognitive domains have not been sufficiently explored (only depression, anxiety and sleep disorders have been individually addressed in greater detail) [5,6,19,20,21]. The mapping of the exact correspondence between NPS and affected cognitive domains is, nonetheless, of crucial importance since it could inform the formulation of individualized preventive strategies directed to the ‘‘fortification’’ of cognitive functions specifically affected by the presence of particular NPS.

Therefore, the aim in undertaking the present study was to explore the relationship between NPS and cognitive performance in CU older (≥60 years) individuals and determine whether specific NPS are related to worse performance in particular cognitive domains. For this purpose, we capitalized on cross-sectional data from the Uniform Data Set (UDS), a standardized set of prospectively collected data on individuals from multiple Alzheimer’s Disease Research Centers (ADRCs) across the United States.

## 2. Materials and Methods

The UDS was initiated in 2005 and continues to be stewarded by the National Alzheimer’s Coordinating Center (NACC). It constitutes the central repository of uniform clinical, neuropathologic and genetic data from all National Institute on Aging/NIH—funded ADRCs across the United States. UDS is freely available to research scientists upon request (https://naccdata.org/). The Institutional Review Boards overseeing each ADRC approved all procedures before the initiation of the study. Informed consent is obtained from all participants or surrogates prior to participation. The rationale, design, data collection process, as well as other key methodological features of the UDS have been described elsewhere [22,23,24]. In brief, data are collected at participating ADRCs by trained physicians and clinic personnel during in-person (office or home) visits or telephone calls, using a common, standardized evaluation protocol. Follow-up assessments are conducted on an approximately annual basis. Data collected are primarily focused on Alzheimer’s disease (AD), although information is also gathered about a variety of other neuropsychiatric disorders.

### 2.1. Eligibility Criteria and Selection of Older Cognitively Unimpaired Participants

The current study was based on data from the baseline (first visit) evaluations of older (≥60 years) CU NACC participants between September 2005 (inception of the UDS) and December 2021 (data freeze) and used data from 43 ADRCs. CU was defined by absence of a cognitive diagnosis of dementia, MCI or cognitive impairment not MCI (see below). Participants being treated with an FDA-approved medication for AD (i.e., tacrine, donepezil, rivastigmine, galantamine and memantine) were excluded as well (receiving such medication challenged the credibility of the physician-based diagnosis). Individuals with a clinician-based diagnosis of a psychiatric disorder (depression, anxiety, schizophrenia, bipolar disorder, post-traumatic stress disorder or other psychiatric conditions) were also excluded (to exclude long-standing neuropsychiatric conditions). Finally, participants receiving antipsychotic, anxiolytic, sedative or hypnotic agents were excluded from the current research due to the potential effect of these medication on neuropsychiatric (improving) and cognitive (impeding) parameters.

CU participants were defined by the absence of dementia, MCI and “cognitive impairment—not MCI,” according to physician-based diagnosis. Cognitive diagnoses were established by either an interdisciplinary consensus team (in the majority of cases) or a single clinician (who conducted the examination), depending on the specific protocol of each ADRC. Diagnoses were based on the personal history, neuropsychological performance, and psychosocial functioning of the participants. MCI and dementia were diagnosed using standard clinical criteria [25,26,27,28,29,30]. Participants with cognitive impairment who did not clearly fit into the categories of CU, MCI, or dementia were diagnosed as cognitively impaired—not MCI.

### 2.2. Measurement of Neuropsychiatric Symptoms

The Neuropsychiatric Inventory Questionnaire (NPI-Q) is a self-administered, widely used tool for the evaluation of NPS in dementia research [31]. NPI-Q evaluates 12 domains namely delusions, hallucinations, agitation/aggression, depression/dysphoria, anxiety, elation/euphoria, apathy/indifference, disinhibition, irritability/lability, aberrant motor behaviour, night-time behaviours, and appetite/eating. Responders are initially requested to report the presence or absence of cardinal symptomatology for each domain in the month preceding the examination and subsequently to rate the severity of these symptoms (if present) according to a 3-point severity scale: mild (noticeable, but not a significant change); moderate (significant, but not a dramatic change); or severe (very marked or prominent; a dramatic change) [32]. For most NPI-Q domains, participants were grouped according to symptoms on a 3-point scale: 0: absent; 1: mild; 2: moderate and severe symptomatology (e.g., irritability: absent, mild, moderate/severe; anxiety: absent, mild, moderate/severe). For the domains of delusions, hallucinations, elation/euphoria and aberrant motor behaviour, owing to the very small number of participants with moderate/severe symptomatology (i.e., from N = 2 with moderate/severe hallucinations to N = 14 with moderate/severe aberrant motor behaviour), participants were dichotomized for presence of these NPS (0: absent; 1: mild, moderate and severe symptomatology).

### 2.3. Measurement of Cognitive Performance

The UDS focuses on the following cognitive domains: episodic memory, semantic memory, language, attention, processing speed and executive function [24]. In the current analysis, episodic memory (immediate and delayed recall) was assessed on the Logical Memory Test (Story A) from the Wechsler Memory Scale—Revised (WMS-R) [33], language on the total word production summing the animal and vegetable fluency tasks [34], semantic memory according to the 30-item version of the Boston Naming Test (BNT-30) [35], attention using the Digit Span Test (DST, forward and backward conditions) from the WMS-R [33], processing speed on the Trail Making Test—Part A (TMT-A) and executive function on the Trail Making Test—Part B (TMT-B) [36]. The administration and scoring of these tests have been previously detailed [24].

### 2.4. Factors and Covariates Considered

Age at the time of the evaluation and education in years of formal schooling were treated as scale variables. Sex, race (Caucasian, African American, American Indian or Alaska Native, Native Hawaiian or Pacific Islander, Asian and multiracial), current use of antidepressant agents and a number of comorbidities that may confound the relationship between neuropsychiatric symptomatology and cognitive performance were treated as categorical variables: history of seizures, traumatic brain injury (TBI), Parkinson’s disease (PD), thyroid disease, vitamin B12 deficiency, alcohol or other substance abuse (with clinically significant impairment occurring over a 12-month period manifested in one of the following areas: work, driving, legal, or social), cardiovascular (heart attack/cardiac arrest or coronal angioplasty/endarterectomy/stent or cardiac bypass procedure or congestive heart failure) and cerebrovascular disease (stroke or transient ischemic attack). These comorbidities were positively assessed according to subject or co-participant reporting of either recent/active or remote/inactive condition. To avoid overadjustment, a statistical criterion was set for the inclusion of each comorbidity in our analysis: only comorbidities that significantly differed between those without and those with one or more NPS were included in our analysis.

### 2.5. Statistical Analysis

Baseline differences between those without and those with one or more NPS were analysed using independent sample *t*-tests (scale variables) and Pearson’s chi-squared tests (categorical variables). Associations between individual NPS and cognitive performance were examined using a multivariate general linear model (GLM). Multivariate GLM is an extension of univariate GLM which deals with multiple continuous dependent variables at once (jointly) while controlling for one or more independent variables. In this way multivariate GLM provides regression analysis and analysis of variance for multiple dependent variables by several factors and covariates while accounting for the potential intercorrelations of the dependent variables and avoiding the pitfalls posed by multiple independent comparisons. A distinct set of regression parameters is generated for each dependent variable.

In the present analysis, six neuropsychological measures were set as the dependent scale parameters: episodic memory [the sum of items recalled in the immediate (0–25 total items recalled) and delayed recall (0–25 total items recalled) tasks], language (the sum of word production in the animal and vegetable 1-min category fluency tasks), semantic memory [BNT-30 (0–30 items retrieved)], attention [the sum of the longest sequences in DST forward (0–8 digits) and backward (0–7 digits) conditions], processing speed [total time in TMT-A (0–150 s)] and executive function [total time in TMT-B (0–300 s)]. The multivariate model featured all 12 NPI-Q variables as independent categorical variables and was adjusted for the factors and covariates described in the previous section (apart from history of seizures and thyroid disease, which did not fulfil the predefined statistical prerequisite, see Table 1). Six functions adhering to the general form of [$y=b_{0}+b_{1}X_{1}+b_{2}X_{2\;}+$…$+b_{i}X_{i}$] were generated, one for each dependent neuropsychological outcome (where *y* = predicted value of the dependent variable for the given values of the independent variables *X*_1−*i*_, *b*_0_ = intercept and *b*_1−*i*_ = regression coefficients, i.e., how much we expect y to change as each independent variable—NPS and confounders—increases).

The statistical analysis was performed using the IBM SPSS Statistics Software Version 27 (Chicago, IL, USA). The conventional threshold of α = 0.05 was implemented for the revelation of statistical significance.

### 2.6. Supplementary Analyses

Apart from the main analysis, two additional sensitivity analyses were performed according to the same general statistical approach. The first one was a confirmatory analysis in which the presence of symptoms for each individual NPI-Q domain was analysed according to a 2-point scale: 0: absent-mild versus 1: moderate-severe symptomatology (a very common practice in dementia research). The exceptions from this rule were (again) the domains of delusions, hallucinations, elation/euphoria and aberrant motor behaviour. The presence of symptoms for these NPS was once again dichotomized as follows: 0: absent; 1: mild, moderate and severe symptomatology (due to the very small number of participants with moderate and severe symptomatology).

The second analysis was exploratory in which every NPS (including delusions, hallucinations, elation/euphoria and aberrant motor behaviour) was analysed according to the following 2-point scale: 0: absent-mild; and 1: moderate-severe symptomatology (despite the formation of very small participant groups with moderate and severe symptomatology).

## 3. Results

### 3.1. Patient Characteristics and Missing Data

The beginning database included 17,605 CU individuals with at least one UDS evaluation. After excluding those younger than 60 years old, individuals with a clinician-based diagnosis of a psychiatric disorder and participants receiving FDA-approved medication for AD, antipsychotic, anxiolytic, sedative or hypnotic agents, a total of 11,882 participants were eligible for the present analyses. Due to the presence of missing data, our adjusted multivariate analysis ultimately involved a total of 7179 older CU participants (Figure 1). The majority of missing data were introduced by the use of alternative neuropsychological tests in the 3rd (last) version of the UDS. Thus, among the total of 11,882 eligible CU participants, there were only 7717, 7712 and 7739 individuals with available neuropsychological assessments according to the Logical Memory Test, BNT-30 and DST.

The baseline characteristics of the included participants are in Table 1 where we compare participants with no NPS and with at least one mild or more severe NPS. Participants were predominantly female, Caucasian, and well-educated. They had a wide range of comorbidities the majority of which were more common in the NPS group (apart from history of seizures and thyroid disease). A total of 1856 individuals had one or more NPS during the month preceding the evaluation: 11 with hallucinations (9 with mild and 2 with moderate), 38 with delusions (29 with mild, 8 with moderate and 1 with severe), 44 with elation/euphoria (31 with mild, 8 with moderate and 5 with severe), 61 with aberrant motor behaviours (47 with mild, 11 with moderate and 3 with severe), 331 with agitation/aggression (252 with mild, 65 with moderate and 14 with severe), 605 with depression/dysphoria (488 with mild, 103 with moderate and 14 with severe), 436 with anxiety (322 with mild, 100 with moderate and 14 with severe), 211 with apathy/indifference (169 with mild, 32 with moderate and 10 with severe), 142 with disinhibition (93 with mild, 31 with moderate and 8 with severe), 658 with irritability/lability (524 with mild, 111 with moderate and 23 with severe), 602 with night-time behaviours (433 with mild, 136 with moderate and 33 with severe) and 332 with appetite/eating disturbances (251 with mild, 68 with moderate and 13 with severe).

### 3.2. The Relationship between Neuropsychiatric Symptoms and Cognitive Performance in Older Cognitively Unimpaired Adults

Table 2 displays the models of the regressions examining associations between individual NPS with neuropsychological performance on the 6 tests we examined. The table displays *betas* with their *p*-values. *Betas* should be interpreted as follows: regarding the trichotomous variables: how much we expect cognitive performance to differ between those without NPS (0) and those with moderate/severe NPS (2), and between those with mild NPS (1) and those with moderate/severe NPS (2); regarding the dichotomous variables: how much we expect cognitive performance to differ between those without NPS (0) and those with NPS (1).

Anxiety, elation, aberrant motor behaviour and hallucinations were the only NPS associated with worse cognitive performance in these CU older individuals (Table 2). Individuals with moderate/severe anxiety performed worse on semantic memory (BNT-30) compared to both participants with mild anxiety (~0.8 fewer correct responses on average) as well as unaffected individuals (~0.95 fewer correct responses). Moreover, those with moderate/severe anxiety scored worse on attention (~0.4 fewer items scored in the DST forward and backward conditions—sum of the longest sequences) compared to non-anxious controls. With respect to executive function, individuals with moderate/severe anxiety required ~10 more seconds to complete the TMT-B task compared to unaffected individuals.

Elation and aberrant motor behaviour affected only one cognitive domain each, attention and language, respectively. Those reporting euphoria scored ~0.8 fewer items in DST forward and backward conditions in comparison with unaffected participants. Regarding motor complains, aberrant motor behaviour was associated with ~2.1 fewer words generated in the category fluency task. Finally, the presence of hallucinations was related to worse performance in processing speed and executive function with affected individuals requiring ~10 and ~43 more seconds than healthy controls to complete the TMT-A and TMT-B tasks, respectively.

The confirmatory analysis (Appendix A) practically reproduced the findings of the main analysis. However, the exploratory analysis provided several inconsistent findings (Appendix A). According to the exploratory analysis, elation was associated with both poorer attention (as in the main analysis) and processing speed while delusions were related to worse processing speed and executive function (motor disturbance was no more associated with SVF and hallucinations were no more linked to processing speed and executive function). The associations of anxiety with semantic memory, attention and executive function remained unaltered.

## 4. Discussion

The present article explored whether specific NPS are related to worse performance on particular cognitive domains in older CU individuals. Our analysis revealed that moderate/severe anxiety was related to worse semantic memory, attention and executive function, hallucinations (moderate/severe delusions in the exploratory analysis) were linked to worse processing speed and executive function, while elation and aberrant motor behaviour were associated with poorer attention and language performance, respectively. The measures of association were relatively small in most cases (except for the relashionship between psychotic symptoms and executive impairment). The remaining NPS were not linked to worse cognitive performance. At this point, it is appropriate to highlight that participants with long-standing neuropsychiatric conditions were excluded from our analysis while NPI-Q focuses on the assessment of NPS that occurred in the month prior to the assessment (ignoring symptoms and behaviours that are usual to the examinee); therefore, the tardive impact of NPS on cognitive performance may have not been captured (our findings probably reflect the earlier cognitive fallout associated with recently occurring NPS).

Late-life anxiety and depression are among the most profoundly studied NPS with respect to cognitive performance. Anxiety has been consistently associated with poorer cognitive performance, and predominantly worse attention and executive functioning [37,38,39]. In fact, anxiety is hypothesized to compromise the very neuropsychological construct of the aforementioned cognitive operations by impeding fixation on task-pertinent stimuli and inhibition of interferences, cognitive flexibility and switching between and within relevant tasks, as well as updating and monitoring of transient information storage [40]. Less often, anxiety has been related to poorer performance in other cognitive functions including semantic memory, but these less consistent associations are more poorly understood and might be a by-product of the generally undermined cognitive functioning [41]. Of note, we failed to reveal an association between depression and cognitive performance. This arguably unexpected finding could be, however, explained by the temporal dynamics of the well-established association between depression and cognitive impairment [16]. Although anxiety is often accompanied by contemporary cognitive ramifications, the overall duration of the disorder and the number of affective episodes appear to be of decisive importance with respect to the effect of depression on cognitive outcomes [42]. As previously mentioned, the tardive impact of NPS on cognitive performance may not have been captured by our study.

Psychotic symptoms, i.e., hallucinations and delusions, are the least frequent NPS among CU individuals [43]. However, they are considered the strongest precursors to cognitive impairment and close surveillance is probably warranted in the presence of late-life psychotic manifestations [6]. Hallucinations and delusions are predominantly associated with incident dementia with Lewy Bodies (DLB) and secondarily with frontotemporal degeneration (FTD) [6]. Both dementia entities share a prominent dysfunction in executive functioning [44] while DLB presents a strong association with attention-processing speed deficits, as well [45]. Intriguingly, our findings suggest that psychotic symptoms may be related to worse executive and processing speed performance in CU older adults (a finding probably consistent with their prominent association with incident DLB and FTD). Although these associations have been reproduced in CU, otherwise healthy older adults [6,46], the majority of relevant research has focused on individuals with psychotic disorders. In this specific subgroup, psychotic symptoms have been predominantly related to executive dysfunction as well, and namely impaired inhibitory control that leads to misinterpretation of experiences or false perceptions in the absence of external stimulation [47,48]. Among the remaining cognitive domains, psychotic disorders have mainly been associated with worse processing speed, which (slower processing) also seems to mediate part of the deleterious effect of psychotic disorders on executive function [49,50,51].

With the exception of sleep disturbances, the remaining NPS have been substantially less investigated with respect to their effect on cognitive outcomes in CU older adults. Contrary to the current study, articles exclusively focusing on the impact of sleep disorders on cognition tend to report a detrimental effect on most cognitive functions [52,53]. However, such articles implement more thorough sleep evaluation protocols separately and meticulously addressing the impact of several crucial sleep parameters and most notably sleep quality, efficiency, duration, and daytime sleepiness, on individual cognitive functions. Therefore, the less rigorous NPI-Q-based assessment of sleep disorders along with the potential temporal dynamics governing the sleep-related effects on cognition may account for the lack of an association between sleep disturbances and cognitive performance in the present analysis.

As for the rest NPS (elation/euphoria, appetite disorders, apathy, irritability, disinhibition, agitation and aberrant motor behaviour), to date, their relationship with cognitive performance has been primarily explored in cognitively impaired populations. However, as indicated by Lü and colleagues, estimated associations vary significantly between individuals with different cognitive backgrounds [54]. Therefore, the extrapolation of these findings to CU individuals would be erroneous. On the aforementioned grounds, future research ought to focus on these understudied NPS to shed more light on their effect on individual cognitive functions in CU older individuals.

### 4.1. Strengths and Limitations

The main strengths of our study are the large sample size of older individuals without cognitive impairment and the rigorous statistical protocol. We were careful to exclude individuals with long-standing psychiatric conditions and/or under treatment with medication targeting neuropsychiatric manifestations while interfering with cognitive performance, as well as to adjust for a number of important confounders. The statistical approach accounted for the potential intercorrelations among the dependent neuropsychological measures while avoided the performance of multiple independent comparisons (minimizing Type I error).

However, it is appropriate to point out that the current analysis presents a number of weaknesses, as well. First, the prevalence of several NPS and most notably of psychotic symptoms was expectedly low. Therefore, as reflected on the low accuracy estimates of the relevant findings, several aspects of our analysis were relatively underpowered. Secondly, NPS were assessed using the NPI-Q, which is a widely used instrument for their evaluation in dementia research. However, the use of more thorough assessment tools (e.g., depression and anxiety scales, sleep quality and efficiency scales, etc.) could potentially be more sensitive in revealing and more accurate in quantifying the severity of NPS. Another limitation of our study stems from its observational, cross-sectional nature, which makes it inappropriate to make any etiologic inferences about NPS and cognitive performance. Therefore, it is important that future longitudinal investigations will complement the findings of this cross-sectional analysis, to better understand the temporal dynamics of NPS and cognitive outcomes. Finally, although we accounted for a number of important confounders, residual confounding cannot be excluded [55,56,57].

### 4.2. Conclusions

In the present study, moderate/severe anxiety was related to worse semantic memory, attention and executive function, psychotic symptoms were associated with worse processing speed and executive function, while elation and aberrant motor behaviour were associated with poorer attention and language performance, respectively. These findings could inform the formulation of individualized preventive strategies directed to the ‘‘fortification’’ of cognitive functions specifically affected by the presence of particular NPS.

## Figures and Tables

**Figure 1 medicina-58-01586-f001:**
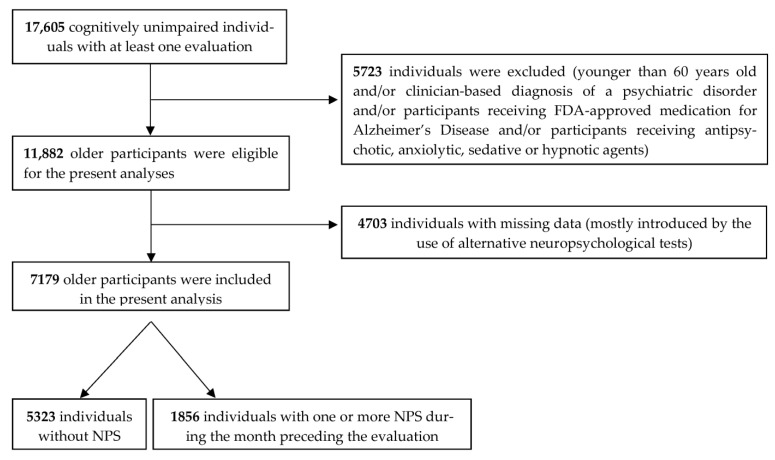
Flowchart of participant selection.

**Table 1 medicina-58-01586-t001:** Baseline differences between cognitively unimpaired individuals without and with one or more neuropsychiatric symptoms.

Variable	Without NPS (n = 5323)	With NPS (n = 1856)	*p*-Value
Age in years	73.63 ± 7.81	73.65 ± 7.87	0.913
Formal education in years	15.72 ± 2.94	15.42 ± 3.11	<0.001
Sex (male/female)	1788 (33.6%)/3535 (66.4%)	762 (41.1%)/1094 (58.9%)	<0.001
Race (Caucasian/African American/American Indian or Alaska Native/Native Hawaiian or Pacific Islander/Asian/Multiracial)	4130 (77.5%)/916 (17.2%)/14 (0.3%)/3 (0.1%)/120 (2.3%)/140 (2.6%)	1580 (85.1%)/190 (10.2%)/8 (0.4%)/1 (0.1%)/22 (2.2%)/55 (3.0%)	<0.001
Cardiovascular disease (No/Yes)	4748 (89.2%)/575 (10.8%)	1608 (86.6%)/248 (13.4%)	0.003
Cerebrovascular disease (No/Yes)	5028 (94.5%)/295 (5.5%)	1706 (91.9%)/150 (8.1%)	<0.001
Parkinson’s disease (No/Yes)	5293 (99.4%)/30 (0.6%)	1810 (97.5%)/46 (2.5%)	<0.001
Traumatic brain injury (No/Yes)	4896 (92.0%)/427 (8.0%)	1662 (89.5%)/194 (10.5%)	0.001
History of seizures (No/Yes)	5237 (98.5%)/82 (1.5%)	1826 (98.5%)/27 (1.5%)	0.798
Thyroid disease (No/Yes)	4324 (81.4%)/990 (18.6%)	1508 (81.3%)/348 (18.7%)	0.909
B12 deficiency (No/Yes)	5146 (96.7%)/177 (3.3%)	1771 (95.4%)/85 (4.6%)	0.013
Alcohol abuse (No/Yes)	5221 (98.1%)/102 (1.9%)	1774 (95.6%)/82 (4.4%)	<0.001
Other substance abuse (No/Yes)	5289 (99.4%)/34 (0.6%)	1835 (98.9%)/21 (1.1%)	0.036
Current antidepressant use (No/Yes)	4844 (91.0%)/479 (9.0%)	1547 (83.4%)/309 (16.6%)	<0.001
Episodic memory (sum of items recalled in the immediate and delayed recall tasks)	25.66 ± 7.71	24.55 ± 7.71	<0.001
Language (sum of word production in the animals and vegetables lists)	34.61 ± 8.57	33.70 ± 8.16	<0.001
Semantic Memory (BNT-30)	26.94 ± 3.41	27.09 ± 3.15	0.104
Attention (sum of longest sequences in DST forward & backward conditions)	11.58 ± 1.99	11.50 ± 1.99	0.139
Processing speed (TMT-A seconds)	35.15 ± 15.27	35.68 ± 17.20	0.212
Executive function (TMT-B seconds)	93.04 ± 50.88	96.25 ± 53.65	0.021

NPS: neuropsychiatric symptoms; BNT: Boston naming test; DST: digit span test; TMT-A: trails making test—part A; TMT-B: trails making test—part B; n = number of participants with available data per parameter.

**Table 2 medicina-58-01586-t002:** Associations between neuropsychiatric manifestations and cognitive performance.

NPS		Episodic Memory		Language		Semantic Memory		Attentio		Processing Speed		Executive Function	
		B	*p*	B	*p*	B	*p*	B	*p*	B	*p*	B	*p*
Depression	0	0.145	0.844	0.398	0.610	−0.382	0.202	0.028	0.887	1.476	0.314	2.080	0.645
	1	−0.165	0.832	−0.018	0.982	−0.140	0.656	0.157	0.445	2.872	0.063	4.381	0.357
	2	Ref		Ref		Ref		Ref		Ref		Ref	
Anxiety	0	1.205	0.100	−0.679	0.379	**0.966 ^(1)^**	**0.001**	**0.387 ^(3)^**	**0.046**	−1.533	0.291	**−10.220 ^(4)^**	**0.022**
	1	0.345	0.670	−1.641	0.055	**0.816 ^(2)^**	**0.013**	0.286	0.183	−1.530	0.341	−6.187	0.211
	2	Ref		Ref		Ref		Ref		Ref		Ref	
Agitation	0	0.437	0.619	−0.099	0.915	−0.323	0.365	0.059	0.802	−0.599	0.731	−0.306	0.955
	1	0.708	0.454	0.079	0.937	−0.168	0.662	0.018	0.944	−3.146	0.094	−7.669	0.185
	2	Ref		Ref		Ref		Ref		Ref		Ref	
Apathy	0	0.922	0.437	1.687	0.178	−0.048	0.920	0.198	0.529	−0.697	0.767	−1.510	0.835
	1	0.283	0.824	0.760	0.573	−0.312	0.547	0.066	0.846	0.780	0.758	2.161	0.782
	2	Ref		Ref		Ref		Ref		Ref		Ref	
Disinhibition	0	0.439	0.724	−0.209	0.874	0.168	0.740	−0.023	0.944	1.012	0.682	−9.822	0.197
	1	−0.285	0.838	−0.413	0.779	−0.375	0.508	0.260	0.482	0.142	0.959	−12.496	0.143
	2	Ref		Ref		Ref		Ref		Ref		Ref	
Irritability	0	0.586	0.403	−0.320	0.665	−0.113	0.690	−0.162	0.384	−2.652	0.056	−2.918	0.495
	1	0.452	0.538	−0.371	0.632	−0.044	0.883	−0.227	0.244	−2.430	0.095	−1.893	0.673
	2	Ref		Ref		Ref		Ref		Ref		Ref	
Night-time	0	0.373	0.519	0.552	0.366	0.134	0.570	0.076	0.621	1.033	0.368	4.938	0.163
	1	−0.292	0.655	0.387	0.575	0.195	0.462	−0.025	0.885	0.789	0.543	6.174	0.122
	2	Ref		Ref		Ref		Ref		Ref		Ref	
Appetite	0	0.233	0.777	1.381	0.112	0.608	0.069	0.137	0.529	−2.717	0.096	−9.708	0.054
	1	−0.336	0.715	1.093	0.260	0.428	0.251	0.089	0.714	−1.296	0.477	−0.913	0.871
	2	Ref		Ref		Ref		Ref		Ref		Ref	
Elation	No	0.565	0.613	0.509	0.666	0.069	0.880	**0.764 ^(5)^**	**0.010**	−2.484	0.263	−3.593	0.599
	Yes	Ref		Ref		Ref		Ref		Ref		Ref	
Motor	No	1.339	0.152	**2.129 ^(6)^**	**0.031**	−0.225	0.554	0.085	0.732	1.752	0.345	−0.173	0.976
	Yes	Ref		Ref		Ref		Ref		Ref		Ref	
Delusions	No	0.866	0.479	0.845	0.513	0.437	0.378	0.415	0.200	−2.535	0.296	−4.524	0.545
	Yes	Ref		Ref		Ref		Ref		Ref		Ref	
Hallucinations	No	3.692	0.094	0.015	0.995	0.324	0.717	0.125	0.830	**−10.272 ^(7)^**	**0.019**	**−43.103 ^(8)^**	**0.001**
	Yes	Ref		Ref		Ref		Ref		Ref		Ref	

**Bold** denotes statistical significance; Ref: reference group; the numbering of the neuropsychiatric symptoms corresponds to: 0 = absent; 1 = mild; 2 = moderate and severe symptomatology; 95% CIs ^(1)^ (0.384, 1.547); ^(2)^ (0.172, 1.460); ^(3)^ (0.007, 0.767); ^(4)^ (−18.980, −1.461); ^(5)^ (0.183, 1.345); ^(6)^ (0.193, 4.065); ^(7)^ (−18.835, −1.710); ^(8)^ (−69.479, −16.727).

## Data Availability

For further information on access to the NACC database, please contact NACC (contact details can be found at https://naccdata.org/).

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
