# Peer review of "The Relationship between Neuropsychiatric Symptoms and Cognitive Performance in Older Adults with Normal Cognition"

_medicina, 2022, doi:10.3390/medicina58111586_

Round 1

Reviewer 1 Report

General comment 

The paper reports on a cross-sectional analysis of the baseline evaluations of cognitively unimpaired elderly participants from the National Alzheimer’s Coordinating Center (NACC) Uniform Data Set (UDS). Some, but not all neuropsychiatric symptoms (NPS) were found to be related to worse performance in specific cognitive domains. The paper is well written, comprehensible not only to specialists, but also to the broad readership of Medicina. The conclusions are supported by the findings and formulated cautiously, under due consideration of the limitations of the study. Considering that NPS often precede cognitive decline in persons developing dementia, the topic clearly is of interest to the field.

Minor issues: 

1.     Abstract, line 10: The abbreviation NPS is used the first time here. It should therefore be preceded by the full term, to read “neuropsychiatric symptoms (NPS)”.

2.     References: As reference #31 the paper on the 10-item version of the NPI (Cummings et al. 1994) is listed. Since the 12-item NPI-Q was used by the ADRCs and data from this version were analyzed, the paper by Kaufer et al. 2000 (J Neuropsychiatry Clin Neurosci 12:2) appears to be a more appropriate reference.

Author Response

Reviewer 1

Comments and Suggestions for Authors

The paper reports on a cross-sectional analysis of the baseline evaluations of cognitively unimpaired elderly participants from the National Alzheimer’s Coordinating Center (NACC) Uniform Data Set (UDS). Some, but not all neuropsychiatric symptoms (NPS) were found to be related to worse performance in specific cognitive domains. The paper is well written, comprehensible not only to specialists, but also to the broad readership of Medicina. The conclusions are supported by the findings and formulated cautiously, under due consideration of the limitations of the study. Considering that NPS often precede cognitive decline in persons developing dementia, the topic clearly is of interest to the field.

Minor issues: 

  1. Abstract, line 10: The abbreviation NPS is used the first time here. It should therefore be preceded by the full term, to read “neuropsychiatric symptoms (NPS)”.
  2. References: As reference #31 the paper on the 10-item version of the NPI (Cummings et al. 1994) is listed. Since the 12-item NPI-Q was used by the ADRCs and data from this version were analyzed, the paper by Kaufer et al. 2000 (J Neuropsychiatry Clin Neurosci 12:2) appears to be a more appropriate reference.

Response: Thank you for your valuable comments and suggestions. ‘‘NPS’’ was defined at first mention in the abstract, and reference 31 was replaced with Kaufer et al., 2000. Thank you for kindly noticing.

Reviewer 2 Report

In this work, authors have explored the relationship between neuropsychiatric symptoms (NPS) and cognitive performance in cognitively unimpaired normal individuals. The idea is to map the correspondence between NPS and different cognitive domains individually to have individualized preventive strategies directed to specific cognitive functions in the presence of any NPS.

Some of the earlier work on NPS discussed in the Discussion section can be brought up in the Introduction.

The materials and methods section is well-written in a detailed manner.

A straightforward statistical approach is used by the authors. The results are not surprising and very ordinary. Hence, the work seems to lack novelty. The supplementary analyses look redundant as you can learn associations between NPS and cognitive performance from the main results. And just organizing the data in different scales doesn’t add any value or give any different information.

We understand the intention of the authors to study the relationship between NPS and cognitive performance in older unimpaired adults. But besides anxiety, there seems to be no significant discovery or knowledge. And that too, the association between anxiety and cognitive functioning is well-studied.

The discussion focuses only on anxiety and hallucination since the results showed a statistically significant association using one statistical tool.

Overall, authors should put efforts into analyzing the data in a more novel and sophisticated way using advanced data mining approaches instead of a standard statistical tool and re-inventing the wheel.

Minor Edits:

Line 10 – Write the full form of NPS since it is used for the first time.

Line 111  - NPI-Q instead of NRI-Q.

Try to arrange Table 2 so that it fits on a single page.

Author Response

Reviewer 2

Comments and Suggestions for Authors

In this work, authors have explored the relationship between neuropsychiatric symptoms (NPS) and cognitive performance in cognitively unimpaired normal individuals. The idea is to map the correspondence between NPS and different cognitive domains individually to have individualized preventive strategies directed to specific cognitive functions in the presence of any NPS.

Response: Thank you for your valuable comments and suggestions. A detailed response is provided below.

Some of the earlier work on NPS discussed in the Discussion section can be brought up in the Introduction. The materials and methods section is well-written in a detailed manner. A straightforward statistical approach is used by the authors. The results are not surprising and very ordinary. Hence, the work seems to lack novelty. The supplementary analyses look redundant as you can learn associations between NPS and cognitive performance from the main results. And just organizing the data in different scales doesn’t add any value or give any different information. We understand the intention of the authors to study the relationship between NPS and cognitive performance in older unimpaired adults. But besides anxiety, there seems to be no significant discovery or knowledge. And that too, the association between anxiety and cognitive functioning is well-studied. The discussion focuses only on anxiety and hallucination since the results showed a statistically significant association using one statistical tool. Overall, authors should put efforts into analyzing the data in a more novel and sophisticated way using advanced data mining approaches instead of a standard statistical tool and re-inventing the wheel.

Response: The reviewer accurately pointed out that the findings of the current report are not very innovative. Although this issue has already been explored, we have reproduced a number of previous findings while rejected several others. The main strengths of our study is the rigorous design and statistical protocol. 1) Individuals with long-standing psychiatric conditions and/or under treatment with medication targeting neuropsychiatric manifestations and interfering with cognitive performance were excluded, a feature which makes the present study rather unique. In this way the present study truly focused on the assessment of NPS that occurred in the month prior to the assessment and our findings reflect early associations between NPS and cognition. 2) Moreover, the statistical approach accounted for the potential intercorrelations among the dependent neuropsychological measures while avoided the performance of multiple independent comparisons (minimizing Type I error). The vast majority (if not the entirety) of published articles implement analytical approaches that do not account for these intercorrelations and tend to associate every newly reported NPS (if it is truly newly reported – see above) with global cognitive changes (relative impairments in practically every cognitive domain). Using a more rigorous approach, we believe that we managed to filter the truly valid associations from the trivial ones.

We also agree with the Reviewer that there is room for more sophisticated analyses, but we hope that the Reviewer understands that such pivotal changes would modify the current paper radically, to the point of practically performing a completely new study. It is our intention however, to take this suggestion into consideration in future studies.

Regarding the supplementary tables, it is true that the re-organization of our data using different scales does not add to the present research. However, different researchers tend to manipulate the NPI-Q in different ways. Therefore, we intended to provide future researchers with the opportunity to capitalize on our findings or reproduce them irrespective of the chosen NPI-Q scaling. This is why this information was added as a supplement. Of course, in case the Reviewer insists, we will remove the supplementary tables.

Minor Edits: 1) Line 10 – Write the full form of NPS since it is used for the first time. 2) Line 111 - NPI-Q instead of NRI-Q. 3) Try to arrange Table 2 so that it fits on a single page.

Response: ‘‘NPS’’ was defined at first mention in the abstract and ‘‘NRI-Q’’ was replaced with NPI-Q. Thank you for kindly noticing. Table 2 was rearranged to fit on a single page.